# Explosive and implosive root concepts: An analysis of music moods rooted by two influential rap artists

**Susumu Nagayama** [1] *, **Hitoshi Mitsuhashi** [2]

1 Center for the Promotion of Social Data Science Education and Research, Hitotsubashi University, Tokyo, Japan, 2 Faculty of Commerce, Waseda University, Tokyo, Japan

* s.nagayama@r.hit-u.ac.jp

## Abstract

This study proposes the notion of "root concepts" in cultural production, defined as a novel style and mode that a creator expresses at the initial field development phase, and that has a great influence on subsequent creators. We explore the role of root concepts in cultural evolution by focusing on their capacity to generate new combinations with other elements and examine how creators use root concepts jointly with other elements. Using data on artists and albums in the rap genre from the online database Allmusic, we view music moods as a type of experience that music generates and focus on music moods as a phenotype in studying styles and modes. We constructed a dataset of recombinatory patterns in the subsequent cultural production and identified two types of root concepts: implosive concepts, which artists use jointly with similar elements; and explosive concepts, which artists use in conjunction with highly diversified elements. Implosive concepts are exclusive because they require creators to have network contagions with those familiar with the root concepts and have strong and specific socio-economic identities. Previous research has suggested that finding a new combination is challenging owing to creators' limited cognitive capacities and the resulting local search. Our finding presents an alternative explanation: some root concepts (i.e., implosive ones) possess innate characteristics that limit creators from experimentally integrating diversified elements. This study develops new opportunities for future research on the evolutionary growth of cultural production and knowledge fields.

## Introduction

Throughout history, innovation has influenced human lives, knowledge, interactions, and understanding of the world. Innovation refers to realized, novel ideas with substantial social, economic, intellectual, or technological impacts [1,2]. Following Schumpeter [3], scholars have employed the recombinatory approach and viewed innovation resulting from recombinations of more than two preexisting elements in knowledge fields [4–6]. Scholars have demonstrated that few ideas are exceptionally novel and influential [7]. Hence, scholars with these insights have investigated how great ideas differ from ordinary ones [8] and how great ideas emerge due to the recombinations of elements [9,10].

**Data Availability Statement:** All data and code files are available from the following Github repository: https://github.com/nagayaman/rootconcept.

**Funding:** S.N..Japan Society for the Promotion of Science KAKENHI Grant Number 19K01923 (https://www.jsps.go.jp/english/). H.M..Japan

Society for the Promotion of Science KAKENHI Grant Number 22H00883 (https://www.jsps.go.jp/english/). H.M.:Japan Society for the Promotion of Science KAKENHI Grant Number 19H01527 (https://www.jsps.go.jp/english/). The funders had no role in study design, data collection and analysis, decision to publish, or preparation of the manuscript.

**Competing interests:** The authors have declared that no competing interests exist.

As a further extension of the literature, scholars have recently started analyzing how great ideas are subsequently used and influence subsequent creators [11]. Research has developed an analytical scheme to measure such subsequent use by the extent to which other works copy the essence of great ideas [12,13]. For instance, using the textual data of speeches during the French Revolution's first parliament, Barron et al. [12] demonstrate that innovative and inspiring speeches are highly different from the preceding speeches but highly similar to the subsequent ones. Using classical piano music notes, Park et al. [13] demonstrate that music styles evolve when artists deviate from previous great composers without losing some of their essences through partial imitation.

We find an additional extension to the literature: while prior studies view great ideas as sheer reference points for imitation to demonstrate their similarity-based influence on subsequent ideas, we know little about how subsequent creators use great ideas jointly with other elements and why they choose such specific elements to recombine with great ideas. Addressing these questions is important because the shift of focus to users of great ideas can reveal a new way of understanding and analyzing great ideas in innovation, provide generalized principles about the social influence of great ideas in innovation, and explain how knowledge fields evolve.

In this study, we propose a notion of root concepts and examine how they are subsequently used and why they are used differently. Root concepts in fields such as music, movies, theatre, cuisine, and writing, refer to novel styles and modes that a creator expresses at the initial field development phase and that have a great influence on subsequent generations of creators. Styles and modes are distinctive and unique patterns of expression. Artists who present root concepts are not necessarily genre founders but those who challenge, destabilize, and alter the status quo of its knowledge fields. Although there are several related concepts, such as breakthrough ideas [14], disruptive ideas [8], or radical innovation [15], we propose the notion of root concepts because *root* captures the evolutionary processes whereby the concepts act as ancestors for developing the "sire lines" of ideas for future generations. Some cultural products are, for example, considered *Picasso-ish* in the case of paintings and *Jimi Hendrix style* in the case of music because they spring from root concepts.

Root concepts are influential because subsequent creators frequently use them in their innovation activities. In doing so, instead of simply copying root concepts, they might recombine them with other selected elements. Other elements to be recombined with root concepts can be diverse or homogeneous. For example, according to AllMusic [16], the Beatles established a novel rock music style and were known to generate many protégés, such as the Carpenters, Pink Floyd, and Oasis. Black Sabbath was heavily involved in the establishment of heavy metal, and they were followed by bands such as Van Halen, Metallica, and Slipknot. Both the Beatles and Black Sabbath generated root concepts with disruptive impacts on the music industry. Many artists in the subsequent generations followed their root concepts, suggesting their influence on later styles. However, those following the Beatles' root concept have used it with highly diversified ideas, whereas those following Black Sabbath's concept have used it with homogeneous ideas. Consequently, products resulting from the recombination of Black Sabbath's root concepts with other homogeneous elements tend to be similar.

As this example demonstrates, root concepts may have different capacity levels to generate new combinations (i.e., generative capacity). Subsequent users may use a root concept jointly with diversified ideas but another root concept only with homogeneous ideas. This study examines this possibility. Moreover, we explore the sources of such differences between root concepts and why root concepts are subsequently used in different manners by focusing on two social factors: social networks and social identity.

We use the data of cultural products by artists and music in the rap/hip-hop genre. We use the term "rap" hereafter to maintain consistency with the AllMusic database label, which is the data source for our analysis. In analyzing the data, we view music moods as a type of experience that music generates. Of the various analytical dimensions in studying styles and modes, we focus on music moods, which fits well with the recombinatory approach that analyzes combinations of elements. We identify two root concepts generated by Run-D.M.C. (hereafter, RD) and N.W.A (hereafter, NWA) in this cultural production field and find that the root concepts of both rap groups are novel and influential. Moreover, subsequent artists have used the root concepts generated by RD with diversified elements and those generated by NWA with homogeneous elements. In other words, their root concepts differ in their capacity to generate new combinations. We call these root concepts explosive and implosive. Explosive root concepts refer to concepts other artists employ with heterogeneous elements, whereas implosive root concepts refer to concepts used with homogeneous elements. We then argue that implosive root concepts require users to have network contagions with those familiar to them (i.e., a contagion requirement) and have specific socio-economic identities (i.e., an identity requirement). Considering these two requirements, the pool of potential elements with which subsequent artists can recombine implosive root concepts may be limited. The findings of our explorative analyses support these arguments.

## Root concepts

### Three approaches to studying great ideas

This study proposes and explores the notion of root concepts to understand how they influence subsequent creators when they are recombined with other elements. Before developing our discussion of root concepts, building a bridge between our arguments and the broader literature would be beneficial. For simplicity, this subsection tentatively employs the term "great ideas" to represent extremely novel and exceptionally influential ideas, such as root concepts.

In studying great ideas, scholars have adopted three approaches. First, some works have articulated the properties of great ideas, which can be as disruptive, in the sense that they disrupt, rather than preserve, streams of ideas if the future ideas inspired by the great ideas do not rely on ideas that the great ideas acknowledge as predecessors [8,17,18]. Great ideas can also be deviant in the sense that they are novel and unique from the stock of ideas from the past [12,13]. In addition, a great idea may have a unique long-term path of influence. Gerow et al. [19] demonstrated that to be intensively cited and become well-recognized, some great ideas require long non-cited periods after their emergence.

Second, some studies have examined how great ideas emerge. Research has demonstrated that emergence stems from recombinations of more than two preexisting elements, noting the challenges and values of connecting seemingly unrelated *dots* in fields [4,5,20–23]. Research has also shown structural, conditional, and demographic factors that promote such emergence, including (but not limited to) brokerage positions in networks [24], constant inflows and outflows of members in knowledge communities [25], creators' long-term time horizons that facilitate explorative search [11], and cognitive diversity in creators' teams [26].

Third, researchers have investigated the impacts of great ideas on the evolution of knowledge fields. Great ideas become reference points for subsequent creators, who (consciously or unconsciously) incorporate some of the essences of great ideas in their creations. Hence, great ideas exert similarity-based influences via a form of imitation [12,13,27]. Some scholars have examined how great ideas persistently influence multiple fields. Using Wikipedia data on global page views of historically well-known people, Yu et al. [28] measured the diversity of influence with the information entropy index. Using forward and backward citation data of

papers from 1900 to 2017, Gates et al. [29] studied the influence of academic papers based on the varieties of disciplines that cite the papers. They find that papers from specific journals or domains have a disproportionately significant influence on multiple domains, and works citing papers from various domains are not necessarily highly cited in various domains.

## Root concepts and generative capacity

This study extends the third thread of the literature by focusing on root concepts in cultural production as a specific form of great ideas and their generative capacity to capture how root concepts differ from each other or are subsequently used. Fink and Reeves [30] present a compelling guide for shaping ideas about generative capacity. Viewing innovation as emerging from an assembly of "components," they argue that some components inherently have a greater capacity to generate new combinations, ultimately causing variations in innovation rates across knowledge fields. In their analysis of how many English words can be generated from letters of the alphabet, they find that ten can be created when using the letters A, B, C, and D; this increases to 28 if E is added to the list; and rises to 46 if F is further added. The findings suggest that some letters as components in combinatory processes have quantitatively different capacities to generate new combinations.

We use their findings as a springboard but qualitatively conceptualize a root concept's generative capacity with the diversity of other elements to be recombined. Subsequent creators may use some root concepts jointly with more highly diversified elements than other root concepts. Considering that diversity is a typical parameter used when studying innovation [29,31], it is reasonable to employ this common parameter as our initial effort. Previous research finds that some knowledge fields can generate a greater number of knowledge recombinations than other fields [29,32]. Similarly, root concepts may exhibit significant heterogeneity in terms of their capacity to *attract* diversified elements, that is, different capacities to generate diversified recombinations.

## Sources of differences in root concepts' generative capacity

If there are differences in the root concepts' generative capacity, it is natural to consider the origin of these differences. Given that root concepts are not overwritten, updated, and time-variant, it is reasonable to attribute them to their innate characteristics. Some root concepts may be exclusive only for specific creators owing to social obstacles that prevent other creators from using the elements in their recombination [33]. Such entry "barriers" limit potential creators, thereby making root concepts exclusive to those with specific attributes or backgrounds. This exclusiveness should lead to a limited variety of elements to be recombined with root concepts.

The entry *barriers* are high because of two requirements imposed on subsequent creators: (1) the contagion requirement, which means that subsequent creators must gain access to the knowledge necessary for using root concepts via network contagion [34–36]; and (2) the identity requirement, whereby subsequent creators must have socio-economic identities that fit with root concepts [33,37].

First, if using some root concepts in innovation activities entails tacit knowledge, creators might need to learn how to use the root concepts via network contagions with those familiar with the concepts [38]. Tacit knowledge tends to be complex and difficult to articulate; moreover, it is often a part of a large system of complex interdependent components [35,39]. Given this complexity, tacit knowledge can be transferred via network contagions: providers and receivers share reference frameworks and cognitive schemes [34]. The contagion requirement suggests that if it is hard to transfer some root concepts without network contagions because

of tacitness, creators as potential users are limited to those in the networks. Those embedded in some specific parts of networks tend to have similar ideas and knowledge, and thus, the varieties of elements that they can use with root concepts must decrease.

Second, some root concepts may require subsequent creators to have specific socio-economic identities. Sociologists, such as Bourdieu [40], have noted the connections between socio-economic identities and cultural production [41,42]. Creators' products reflect and reproduce their socio-economic identity [43]. Notably, not all cultural products equally function as symbols and identification carriers; thus, the extent of the reflection varies across creators or their products [44]. If some root concepts are compatible only with specific and strong socio-economic identities and are highly exclusive for those not members of certain social groups, then the range of creators as potential users must be rather limited.

Hence, the two requirements for potential users may limit some root concepts' capacity to generate combinations with more diversified elements. We view the two requirements as not independent of each other. The principle of homophily suggests that those with similar socio-economic identities are more likely to interact with each other because the similarities create shared frames of reference and present social space for interactions [45–47].

## Research context

### Data

We analyzed the data of artists and musicians in the rap genre. Rap is music with rhyming lyrics, where vocalists use spoken language and chant with insistent beats or musical accompaniments [48]. The half-century-old genre is an ideal choice because it has neither an extremely long history that could hamper analysis to discern root concepts nor an extremely short history that may present challenges in assessing root concepts' long-term influences on subsequent innovation activities. Moreover, the study can employ the mood artists express via music as a phenotype of styles and modes [49]. This genre started as local party music in the New York suburbs of Harlem and the Bronx in the 1970s, and early artists include DJ Kool Herc, Afrika Bambaataa, and Grandmaster Flash [50,51]. The Sugar Hill Gang and the Fatback Band released mainstream records from the early 1970s to the early 1980s, followed by a period commonly known as the Golden Age, which established rap music as a commercial market.

Studies adopting the recombinatory approach hinge on the phenotype of knowledge. For example, in their analysis of the patent application data, Fleming et al. [4] viewed patent subclasses as phenotypes for studying combinations and measured the novelty of an innovator's technological and intellectual knowledge with the number of new subclass combinations in each of their patent applications. As another example, in their analysis of innovation in the fashion industry, Godart et al. [52] decomposed clothing styles into color, fabric, print, pattern, and look and examined how fashion houses adopt combinations of these elements.

Of the other key components of music, such as melodies, harmonies, and rhythm, we view music moods as phenotypic elements to be recombined. Music generates moods and interactions between artists and audiences. Indeed, previous research used music moods as a phenotype in styles and modes [49,53–55]. In a seminal work, Hevner [54] studied the expressiveness of music through the analysis of affective values that music generates, such as joy, dignity, and dreaminess. Spotify also supports this view of moods as one of the key components by developing technology to draw inferences about audiences' moods based on their voices and suggest songs that match emotional states [56].

By recombining some moods, artists generate and deliver specific core moods. In rap music, artists create coherent and complex sets of melodies (or flows), harmonies, rhythms, and lyrics to create moods, such as anger, playful, trippy, stylish, confrontational, energetic,

and toughness. Given that artists embed multiple moods in music, we can adopt the recombinatory approach to assess artists' novelty by using moods as a phenotype and studying the combinations of moods that they express in their music. Other elements of rap music may be important representations of influence from others, including lyrical and musical sampling, song speed, beats, samplings, and lyrics [57]. However, focusing on these elements would pose empirical challenges, as the recombinatory approach requires that specifications of musical elements be recombined. Hence, using moods as a phenotype is analytically convenient for assessing artists' recombinatory patterns. Nonetheless, we understand the importance of lyrics in this genre and use some lyrical data in our supplementary analysis regarding the contagion requirement.

We collected our data mainly from AllMusic [16], which is a reliable online music database that provides basic information on albums and artists, including the release date, genres, music styles, song titles, and credits. It obtains data from a company called TiVo. Unlike crowdsourcing sites, such as Discogs or MusicBrainz, TiVo collects music metadata from internal experts and outsourced freelancers and supplies the data to data-mining businesses at firms such as Google and Microsoft [58,59].

A unique advantage of this database is the data available regarding the types of moods expressed by artists in their music. Additionally, it offers *followed by* data (i.e., by whom a given artist is followed) and *influenced by* data (i.e., who influences a given artist), which we can use to measure influences and construct *mentor-protégé* networks. However, two shortcomings of the database are notable: (1) the extent of sample representativeness is unknown, and (2) the uniqueness of this dataset makes it difficult for us to conduct cross-validation for checking the data quality.

We collected data from 6,286 rap albums released from 1970 to 2013, comprising 3,143 unique artists or groups. Given that the origin of the rap genre has been debated, we adopted a conservative approach and collected the earliest data in the database.

## Three phases of analyses and results

To address our research question, we adopted the following three analytical steps in an explorative way. First, we identified artists who have created root concepts in the genre of rap music. Second, to examine variations in root concepts' generative capacity, we investigated how artists in the subsequent generation use root concepts and how they combine these concepts with other ideas for their music. Finally, we explored the sources of these differences and assessed how the contagion and identity requirements are associated with the use of root concepts by artists in the subsequent generations.

## Step 1: Finding root concepts

We start our analysis by asking two questions: (1) who generated root concepts in the early history of rap music? and (2) what are the phenotypes of their root concepts? For an idea to be a root concept, it must be exceptionally (i) influential and (ii) novel. To measure artist influence, we used the aforementioned data of *followed by* and *influenced by*. Combining these datasets, we construct an adjacent directional influence network matrix and then compute in-degree and PageRank centrality [60].

In-degree centrality indicates the number of other artists $j$ who follow the focal artist $i$. That is,

$$Degree_i = \sum_j A_{ij},$$ (1)

where *A* is a directed adjacent matrix of artists' influence, in which $A_{ij}$ equals 1 if an artist *i* influences an artist *j* (i.e., *i* is followed by *j*) and 0 otherwise.

Similarly, PageRank centrality increases when the focal artist influences another artist who is also influential in the field. The formula of PageRank centrality is as follows:

$$PageRank_i = 1 + \alpha \sum_j \frac{A_{ij}PageRank_j}{\sum_k A_{kj}}. \tag{2}$$

To compute the scores, we use *igraph* for R [61] and set the value of α equal to .85, which is suggested by Page et al. [60].

To measure artists' novelty, we follow Fleming et al. [4] and consider an idea to be novel when an artist expresses mood combinations in the music that no other artists have ever developed prior to their music. For each artist in our sample, we create the variable of a novel mood pair by (1) compiling all the data on moods expressed in albums, (2) constructing the data of all dyads of moods drawn from the compiled sets, and (3) counting the number of dyads not previously present. Suppose that $m_i$ is a music mood, and $\Omega_k$ is a set of album *k*'s moods. $|\Omega_k|$ counts the number of moods presented in album *k* (i.e., cardinality). $2^{\Omega_k}$ is then a powerset of unordered mood pairs drawn from $\Omega_k$, and $\bigcup_q 2^{\Omega_q}$ is the union of the mood pair powersets that consist of album 1 to *q*. We measure album *k*'s novelty with:

$$Novelty_k = |2^{\Omega_k} - \bigcup_q 2^{\Omega_q}|, \tag{3}$$

where *q* is all albums in the genre of rap music released before album *k*.

With these three measures, we construct the 3D scatter plots in Fig 1, where we use in-degree centrality, PageRank centrality, and novel mood pair as the x-, y-, and z-axis,

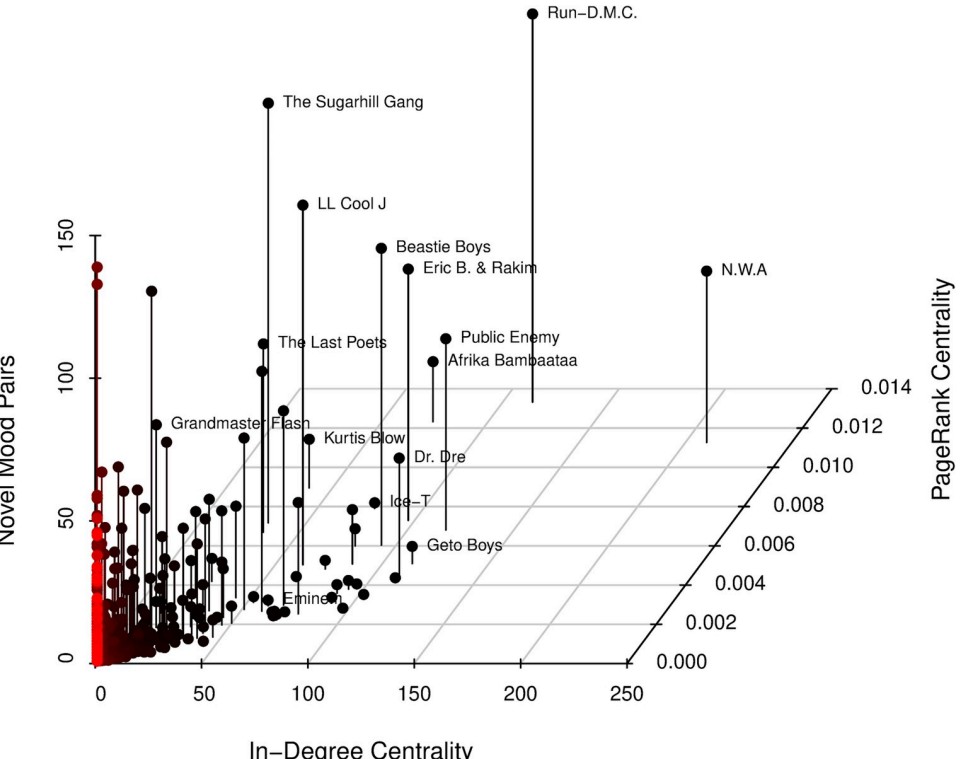

**Fig 1. 3D scatter plot of influence and novelty of rappers' first albums (N = 1,154).**

respectively. S1 Table reports the descriptive statistics of these measures; the dots in the figure represent artists, and labels are added to famous artists with high scores. Our visual inspection suggests that RD and NWA are influential, as measured with in-degree and PageRank centrality. Moreover, they are novel, measured with combinatory novelty. Using the standardized data, we calculate the means of in-degree centrality and PageRank scores and then multiply the means with the novelty scores. We confirm that out of 1,154 artists, RD and NWA are ranked first and second in this single score distribution, respectively. We also plot the cumulative population-level distributions of the three measures, calculate the extremeness of their values in the distributions, and ensure that the two artists are uniquely different from others (see S1 Fig).

Furthermore, RD and NWA are influential because of their commercial success and visibility. RD is an East Coast-based rap group that released its first studio album in 1984 and established the New School genre. The group became the first rap act with a gold record and appeared on the cover of *Rolling Stone* and was ranked 48th in *Rolling Stone*'s 100 Greatest Artists [62]. Experts acknowledge that RD attained great commercial success and significantly influenced several well-known artists, including Public Enemy, LL Cool J, and the Beastie Boys [51]. In contrast to RD, NWA was a West Coast-based rap group that released its first studio album in 1988. Like RD, NWA also achieved great commercial success and established Gangsta rap. Dr. Dre, an NWA member, was ranked 56th in *Rolling Stone*'s 100 Greatest Artists list [62], and NWA significantly influenced several big rap stars, such as Snoop Dogg, 2Pac, and Eminem [51].

To understand the phenotypes of the root concepts, we create the vectors of moods for all albums released between 1970 and 2013 in the rap music genre. The vectors can contain up to 272 moods and have 0 or 1 values. An element is coded 1 if a given album contains the corresponding moods. On average, the number of moods per album is 11.90. Some moods, such as anger and outrage, are linguistically related; thus, to find core moods, we simplify the data by running the topic modeling of Latent Dirichlet Allocation [63] in *topicmodels* for R [64].

Considering that the fitness indexes [65,66] are highest (or lowest) when we extract four topics (i.e., core moods) from the data, we extract the following topics: (1) tough, (2) energetic, (3) introspective, and (4) violent. Table 1 presents mood examples for each topic. With these four topics, we examine what moods the phenotypes of their root concepts RD and NWA expressed in their first albums, and the results are summarized below in Table 1. We find that RD and NWA similarly expressed the moods *introspective* and *tough*; however, RD uniquely expressed the mood *energetic*, and NWA uniquely expressed the mood *violent*.

To ensure the significance of these differences, we compare the Euclidean RD-NWA distance scores with the means of the distance scores between all pairs of any two albums in our sample. To compute the distance scores, we use the following equation:

$$Distance_{kl} = \frac{1}{n(n-1)} \sum_{k \neq l} \| A_k - A_l \| = \frac{1}{n(n-1)} \sum_{k \neq l} \sqrt{\sum_{i=1}^{4} (a_{ki} - a_{li})^2}, \qquad (4)$$

where $A_k$ ($A_l$) represents the mood topic vector of album $k(l)$, $a_{ki}$ ($a_{li}$) is an element $i$ of topic probability score in $A_k$ ($A_l$), and $n$ represents the number of albums for comparison.

We first compute the Euclidean distance of topic scores between RD's and NWA's albums and obtain 0.18. We then compute the Euclidean distance of topic scores between all pairs of albums drawn from the population (n = 19,753,756) and find a mean distance of 0.11. The Euclidean distance between RD and NWA is ranked in the top 10.4%. The probability of the population average being 0.18 is $2 \times 10^{-16}$ ($|t| = 6045.00$) (see S2 Fig). Hence, we conclude that

Table 1. Root concepts and core moods.

| Root concepts and corresponding moods | Topic 1 Tough | Topic 2 Energetic | Topic 3 Introspective | Topic 4 Violent |
|---|---|---|---|---|
| **Top ten associated moods** | Street-smart Brash Tough Confident Aggressive Bravado Hedonistic Sleazy Rowdy Swaggering | Energetic Celebratory Rousing Boisterous Confident Freewheeling Playful Fun Rambunctious Exuberant | Stylish Dramatic Literate Reflective Laidback-mellow Nocturnal Druggy Ambitious Cerebral Passionate | Confrontational Menacing Aggressive Rebellious Provocative Intense Harsh Hostile Volatile Uncompromising |
| *Run-D.M.C.* by Run-D.M.C. <br> aggressive, boisterous, brash, bravado, celebratory, confident, confrontational, energetic, exuberant, fiery, freewheeling, fun, gritty, irreverent, playful, rambunctious, rebellious, reckless, rousing, street-smart, swaggering, tough, uncompromising, urgent, visceral, and witty | 0.26 | **0.31** | 0.18 | 0.26 |
| *Straight Outta Compton* by N.W.A <br> aggressive, angry, boisterous, brash, cathartic, confident, confrontational, fiery, harsh, hedonistic, hostile, humorous, intense, malevolent, menacing, outrageous, provocative, rambunctious, raucous, rebellious, reckless, rowdy, street-smart, tense/anxious, tough, unsettling, and volatile | 0.24 | 0.20 | 0.17 | **0.39** |

Notes: Each weight for the four topics is calculated based on the estimated topic model. Higher scores indicate stronger associations with the corresponding topics. The table illustrates that while Run-D.M.C.'s core mood is energetic, N.W.A's core mood is violent.

the unique phenotypic characteristics of the two groups' root concepts are energetic and violent, respectively.

## Step 2: Subsequent artists' use of root concepts

This subsection examines how the two artists' root concepts are subsequently used and how they differ in terms of the capacity to generate new combinations using a regression analysis. We assess the characteristics of their root concepts by focusing on the diversity of elements to be recombined. In doing so, we use the following three diversity measures as our dependent variables: *combined elements*, *element diversity*, and *combinatory strength*. Let $m_{it}$ be a mood $i$ in year $t$, and $M_t$ be an adjacent undirected network matrix of mood combinations in year $t$. We weight $(m_{it}, m_{jt})$ in $M_t$ with the frequency of $m_{it}$–$m_{jt}$ combinations in year $t$. As we noted above, $\Omega_k$ is a set of album $k$'s moods, and $2^{\Omega_k}$ is a powerset of unordered mood pairs drawn from $\Omega_k$.

First, *combined elements* refer to the unique number of other moods to be combined with the focal mood $i$ in year $t$:

$$Combined\ Elements_{it} = \left| \left\{ \{m_{it}, x\} | \{m_{it}, x\} \in \bigcup_k 2^{\Omega_k} \right\} \right|, \tag{5}$$

where $k$ is all albums released in a given year, and $x$ is a mood element jointly used with the given mood $m_{it}$. A higher score indicates that the focal mood $i$ is combined with more diversified moods.

Second, *element diversity* refers to the degree of diversity of moods to be combined based on Blau's heterogeneity index [67]:

$$Element\ Diversity_{it} = 1 - \sum_j P(M_{ijt})^2, \tag{6}$$

where $M_{ijt}$ is the number of $(m_{it}, m_{jt})$ combinations in year $t$, and $P(M_{ijt})$ is $M_{ijt}$ divided by the

total number of mood $i$'s usage in year $t$ (i.e., the probability of the $(m_{it}, m_{jt})$ combinations). A higher score of element diversity indicates that mood $i$ is combined with more diversified moods.

Third, *combinatory strength* captures the strength of ties of $m_{it}$ with the combined elements. We counted the total number of times $m_{it}$ is combined with other moods and divided it by the number of combined moods:

$$Combinatory\ Strength_{it} = \frac{\sum_j M_{ijt}}{Combined\ Elements_{it}}. \tag{7}$$

For example, focal $m_i$ is combined with $m_1$, $m_2$, and $m_3$ 100 times in year t. The $m_{it}$'s combinatory strength is then 100/3. If $m_{it}$ is combined with fewer moods, it has stronger *ties* with specific moods. By definition, element diversity decreases with an increase in its combinatory strength.

The independent variables are binary variables coded as 1 if RD or NWA expresses a given mood in their music and 0 otherwise. If RD expresses a given mood, this binary variable is equal to one. The larger the effect size of this RD variable on combined elements and element diversity, the more it is likely that RD's mood is being used jointly with many different elements. By contrast, the smaller the effect size of this binary variable on combinatory strength, the more likely that RD's mood is repeatedly used with specific elements. The linear model for estimating the effects on combined elements, element diversity, and combinatory strength is as follows:

$$y_{it} = \beta_0 + \beta_1 RD_{it} + \beta_2 NWA_{it} + \sum_j \gamma_j x_{jit} + \varepsilon_{it}, \tag{8}$$

where $RD_{it}$ and $NWA_{it}$ indicate binary independent variables, $x_{1it} \ldots x_{jit}$ indicate control variables, and $\varepsilon_{it}$ is an error term. In Eq (8), we use the following control variables. *Mood age* is the number of years since the focal mood first appeared in the data. *Mood popularity* is the number of albums in which artists use focal moods in a given year; the *total number of moods* is the population-level number of unique moods in a given year; and the *total number of released albums* is the number of albums released in a given year. The S2 and S3 Tables report the descriptive statistics of all variables in the model and the full results of the regression analysis.

To estimate the effects on the aforementioned three diversity measures, we follow previous research taking the recombinatory approach [52] and run regression models with the mood-year as the unit of analysis. We construct mood-year panel data with 4,040 observations from 1989 to 2013. To measure the subsequent influence of RD and NWA, we use the year 1989, one year after NWA's first album was released, as the starting data point. We use generalized estimating equations to account for the intra-correlations in repeated observations for the same moods over time [68]. To ease interpretations, the three diversity measures are scaled.

Fig 2 illustrates the estimated coefficients of the two main independent variables, RD's and NWA's moods, from the results of generalized estimating equations regressions. Fig 2A illustrates that a mood's combined elements are greater if the mood is expressed by RD ($\beta = 0.28$, $p = 4.5 \cdot 10^{-5}$) rather than NWA ($\beta = -0.01$, $p = 0.92$). Fig 2B illustrates that a mood's element diversity is greater if expressed by RD ($\beta = 0.17$, $p = 2.4 \cdot 10^{-5}$) rather than by NWA ($\beta = 0.12$, $p = 0.002$). Moreover, Fig 2C illustrates that a mood's combinatory strength is greater if expressed by NWA ($\beta = 0.23$, $p = 1.5 \cdot 10^{-7}$) rather than by RD ($\beta = 0.14$, $p = 3.8 \cdot 10^{-15}$). Given that a mood's combinatory strength should be smaller if its element diversity is greater, the findings are consistent.

The findings suggest variations in patterns by which other artists use the two root concepts and connote them. Artists in the subsequent generations use RD's root concepts jointly with

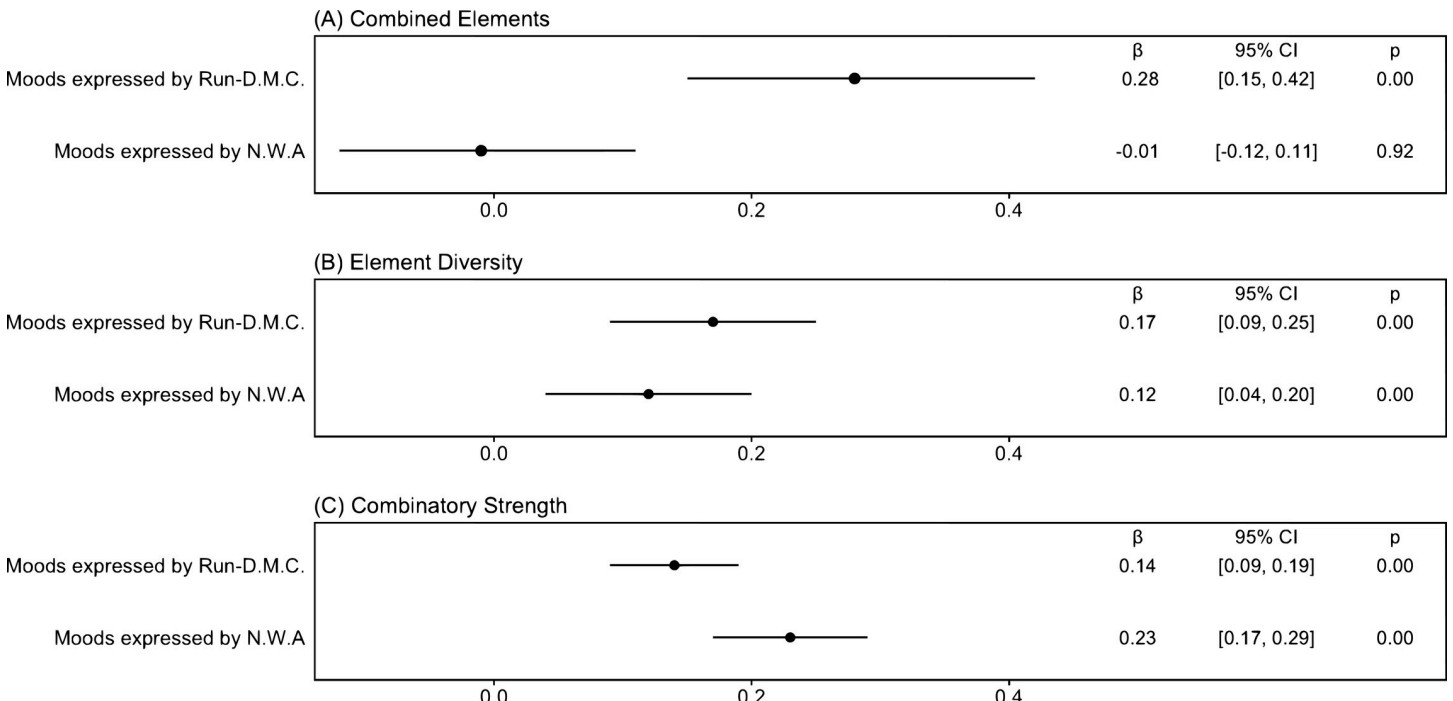

**Fig 2. Coefficient plots—A mood expressed by Run-D.M.C. has greater combined elements, element diversity, and lower combinatory strength than a mood expressed by N.W.A (N = 4,040).** The figure includes point estimates and 95% confidence intervals. We report the descriptive statistics of variables used in the model and regression results in the S2 and S3 Tables.

other dissimilar and highly diversified moods. Such root concepts attract a set of diversified elements to be recombined, which we call "explosive root concepts" because the influence evolves with unlimited scope.

By contrast, artists in the subsequent generations use NWA's root concepts jointly with other similar moods. The recombined elements are homogeneous; thus, we call this type "implosive root concepts."

Both are persistently influential over time, with different evolutionary trajectories. One is subsequently joined with heterogeneous elements, whereas the other is joined with homogeneous elements. We adopt the term "implosive" from atomic bomb engineering. An implosive bomb explodes when explosives (surrounding a spherical case in the center of which plutonium is placed) are used to create inward pressure on the core. The term "implosive root concepts" captures the size and constrained inward direction of the influences of the root concepts on artists in the subsequent generations.

## Step 3: Sources of differences

The analysis so far demonstrates that RD presents an explosive root concept that has joined with a wide variety of other elements (i.e., moods), whereas NWA's implosive root concept has limited generative capacity in the sense that it has recombined only with homogeneous elements. This subsection further explores the source of these differences and assesses our working hypotheses about the contagion and identity requirements.

**The contagion requirement.** The contagion requirement suggests that creators in subsequent generations recombine implosive root concepts only with homogeneous elements because knowledge regarding how to use root concepts is tacit and can be transferred via

network contacts [34]. Considering that those with network contacts tend to have similar social backgrounds and frames of reference [46], the users of implosive root concepts tend to be homogeneous and have similar ideas.

We assess the contagion requirement by testing the possibility that artists in the subsequent generations express NWA-like moods when working with others who used to collaborate with NWA (i.e., NWA's collaborators). By contrast, this network effect should be weaker if artists work with RD's collaborators. We create a binary variable coded as 1 if a given artist worked with NWA's (or RD's) collaborators in making the analyzed album. We focus on the collaborators of NWA's (n = 56) and RD's first three albums (n = 29). The dependent variable captures the extent to which moods expressed by the given artist are proximate to NWA's (or RD's) root concepts. We calculate the cosine similarity scores between the mood topic vector of the given artist's album and NWA's (or RD's) first album, which can be formally expressed as:

$$Similarity_k = \frac{A_R \cdot A_k}{\| A_R \| \| A_l \|} = \frac{\sum_1^4 A_{Ri} A_{ki}}{\sqrt{\sum_1^4 A_{Ri}^2} \sqrt{\sum_1^4 A_{ki}^2}}, \tag{9}$$

where $A_R$ is the mood topic vector of NWA's album (or RD's) and $A_k$ is that of the given album. We construct a dataset of 6,111 album-level observations and use ordinary least squares regression models with robust standard errors. The estimation model is as follows:

$$y_i = \beta_0 + \beta_1 RD_i + \beta_2 NWA_i + \sum_j \gamma_j x_{ji} + \varepsilon_i. \tag{10}$$

The model predicts the effects of the collaboration variables ($RD_i$ and $NWA_i$) on focal album $i$'s mood similarity with RD and NWA after controlling the effects of $x_{ji}$. As control variables, $x_{ji}$, we include location NY/CA, which is a binary variable that captures whether an artist originates from or releases an album from a label headquartered in New York or California. French [69] notes that geographic locations influence artists' music production, and we include those two states as the initial centers of rap music. We collect these location data from the address of the artist's label from Discogs, a music database website. We also include previous releases, referring to the population-level number of albums previously released in the rap genre. Team size is the number of individuals listed in the credit section of a given album. We also include a binary variable indicating artists' gang affiliations and atypical production resources of their albums (see below) and the five-year cohort fixed effects.

Fig 3A and 3B demonstrate the results of regression analyses for testing the contagion requirement hypothesis. We find that an album of an artist who works with RD's collaborators is not associated with the expression of RD-like moods ($\beta$ = −0.07, $p$ = 0.37), whereas an album of an artist who works with NWA's collaborators is pertinent to the expression of NWA-like moods ($\beta$ = 0.37, $p$ = 6.4·10$^{-9}$). The results indicate that network contagion, such as working with NWA's collaborators, is a requirement for an artist to express the NWA-like moods, whereas network contagion is not necessary for an artist to express the RD-like moods.

**Validating the contagion requirement.** The contagion requirement rests on the following premise: knowledge regarding how to use implosive root concepts is tacit and, thus, can be transferred only via network contagion [34,35]. To validate this premise, we conduct the following additional analyses. First, we analyze the differences in lyric tacitness between the two root concepts. Given that tacit knowledge is tied to activities [38], we first measure tacitness by how densely RD and NWA pack words into a second without disrupting rhythms or losing the meaning of their stories to deliver moods. We collect the lyrics and duration data of songs from the first three albums by RD (n = 31) and NWA (n = 37) and divide the number of words in their songs by the song duration. Fig 4A presents the results; it demonstrates that NWA

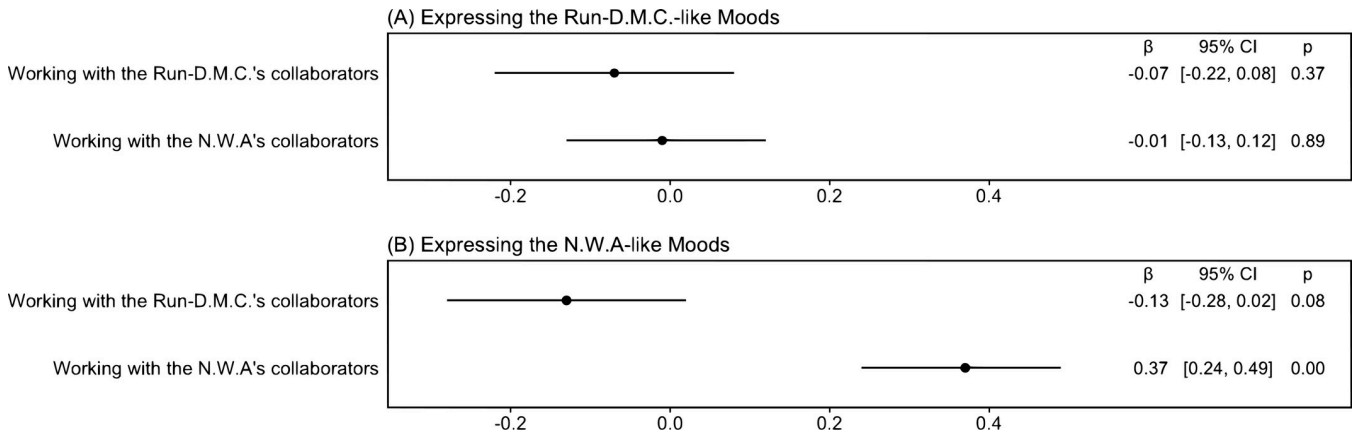

**Fig 3. Coefficient plots—In an album, an artist who works with N.W.A collaborators tends to express moods similar to those expressed by N.W.A (N = 6,111).** The figures include point estimates and 95% confidence intervals. We report the descriptive statistics of variables used in the model and regression results in the S4 and S5 Tables.

uses more words per second than RD, and the difference between them (= 1.95–2.80) is statistically significant ($p = 8.3 \cdot 10^{-6}$).

Second, we adopt an alternative measure of tacitness by focusing on the use of informal words in songs. Using informal words is an important activity in rap music because some slang and interjections can create a rap-unique groove and rhymes [70]. Thus, knowledge for using more informal words without disrupting meaning entails high tacitness. We use the Linguistic Inquiry and Word Count dictionary of informal words and count the number of informal words per second by RD and NWA [71], and Fig 4B illustrates the results. The number of informal words per second is 0.04 for RD and 0.24 for NWA ($p = 6.3 \cdot 10^{-11}$). Hence, it is fair to conclude that knowledge embedded in NWA's implosive root concepts is more tacit than that embedded in RD's root concepts.

Third, to directly assess the working hypothesis that knowledge for implosive root concepts can be transferred via network contagions, we re-analyze the directional networks of influence by resorting to Albert and Barabási's node removal method [72]. An underlying principle of

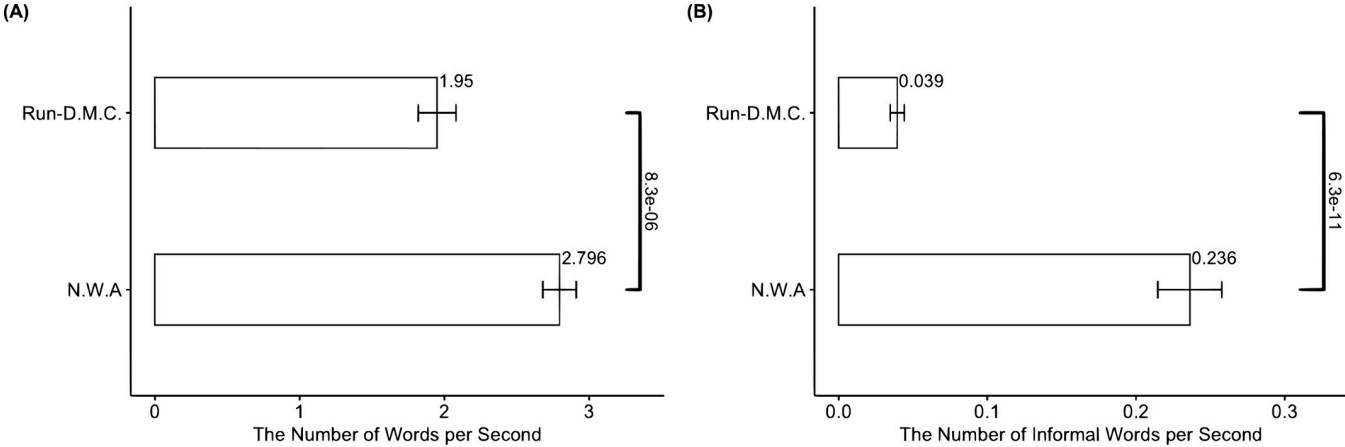

**Fig 4. Differences in root concepts' lyric tacitness.** Panel (A) indicates the means of the number of words in lyrics divided by song duration for RD and NWA. Panel (B) indicates the means of the number of informal words per second. We report the *p*-values from the t-test between the two groups. The error bars in the panels represent standard errors.

their method is that to detect the degree of influence that some specific nodes exert on the overall structures of networks, one should remove them from networks and assess how the node removals cause changes in structural measures of networks in the remaining nodes. Following this idea, we predict that removing artists working with RD (or NWA) from the influence networks would yield significant changes in the remaining network structures. Changes in the influence networks can be best captured with measures that represent nodes' reachability in networks, including (1) the largest path length, (2) average path lengths, and (3) the number of communities in networks:

$$Largest\ Path\ Length_G = \max_{v_i, v_j \in V} d(v_i, v_j), \tag{11}$$

$$Average\ Path\ Length_G = \frac{1}{n \cdot (n-1)} \sum_{i \neq j} d(v_i, v_j),\ and \tag{12}$$

$$Communities_G = \sum_p C_p, \tag{13}$$

where $G$ denotes an adjacent directional influence network matrix after random (or specific) node removals; $V$ denotes a set of entire rap artists except for those removed, and $n$ is the number of artists in $V$; $d(v_i, v_j)$ represents the number of steps of the shortest path from node $i$ to $j$ (i.e., from artist $i$ to $j$); and $C$ indicates a community that we extracted from $G$ with Louvain method [73]. This analysis should alleviate concern about reverse causality: while our regression models suggest that those working with the NWA's collaborators express NWA-like moods, artists wishing to express NWA-like moods may invite NWA's collaborators to work with them.

Suppose the influences of implosive root concepts are based on network contagions. Node removals would then cause a large cluster of nodes at the center of the influence networks to disappear, resulting in a greater number of divided communities, which would reduce the remaining nodes' reachability to other nodes. By contrast, in the case of explosive root concepts, holes at the core, which the node removals create, should be smaller owing to weak associations between influence and network contagions (see S3 Fig). If only implosive root concepts have contagion requirements, and if we remove those working with the collaborators of NWA rather than the collaborators of RD, we expect changes in the network measures to be greater.

With this expectation, we assess changes in network measures after we remove the collaborators of RD (n = 146) and NWA (n = 162). Considering that the observed changes may simply reflect network size changes, we randomly remove 146 and 162 nodes from the networks 10,000 times and obtain the post-removal measures. Hence, we calculate three measures for the following five patterns: (1) no removal, (2) random removals of 146 nodes, (3) random removals of 162 nodes, (4) removals of nodes that worked with RD's collaborators, and (5) removals of nodes that worked with NWA's collaborators. Fig 5 presents the results.

Removing nodes that worked with NWA's collaborators substantially disrupts the structures of the influence networks. The removals increase the path lengths of the remaining nodes (panels (A) and (B)) and generate more communities (panel (C)). Changes in these measures are greater when we remove artists working with NWA's collaborators from the influence networks than those randomly drawn and those working with RD's collaborators. The node removal causes the collapse of the influence networks, suggesting that NWA's influence and network contagions are closely related.

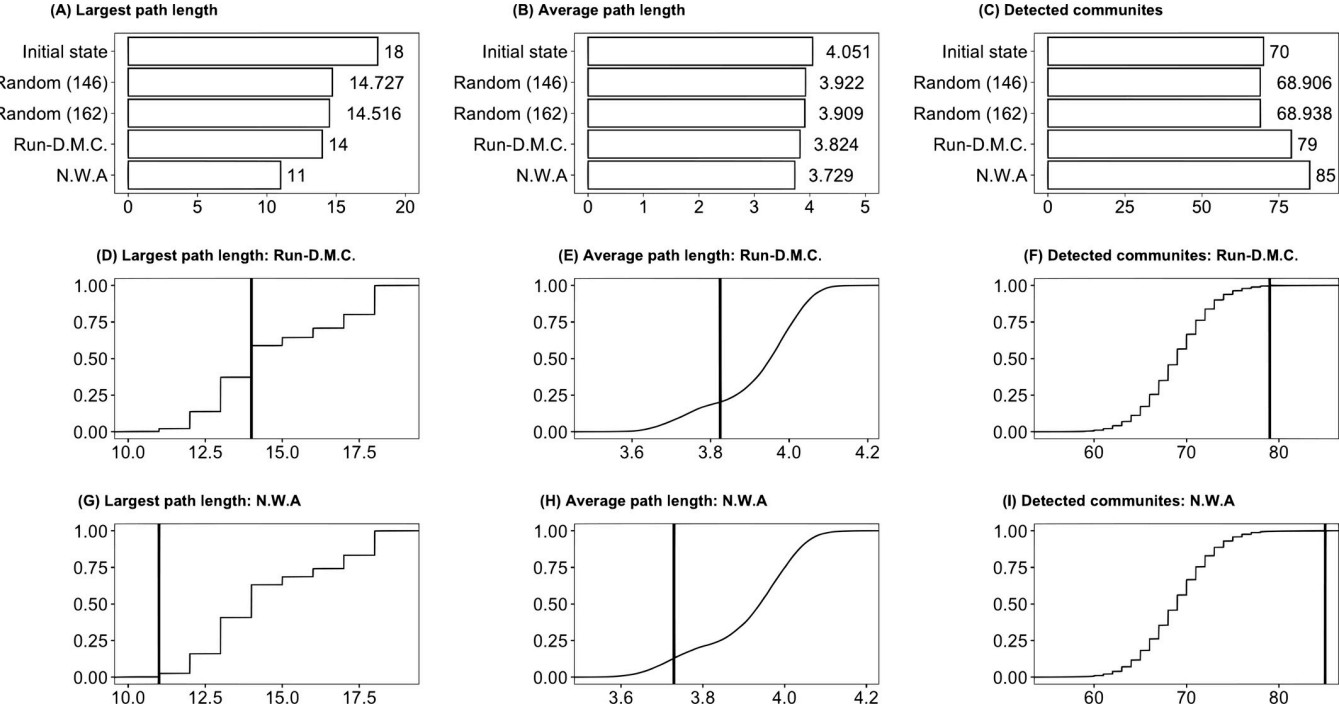

**Fig 5. Effects of node removal for analyzing contagion-influence associations.** We measure three indicators to observe the effects of node removals from the influence networks. (1) largest path length (panels (A), (D), and (G)); (2) average path length (panels (B), (E), and (H)); and (3) the number of communities detected by the Louvain method (panels (C), (F), and (I)). The numbers of artists who worked with RD's and NWA's collaborators are 146 and 162, respectively. Thus, we randomly remove 146 and 162 nodes 10,000 times and report the averages. In the second and third rows, we present the cumulative distributions of results from the randomly removed samples and add the vertical red lines to show the corresponding data for RD and NWA.

The differences caused by node removal can be trivial; thus, we examine how the generated differences are probabilistically rare relative to the randomly generated differences in panels of Fig 5. The figure illustrates the cumulative distributions of changes in the three measures when we conduct random removals 10,000 times. In panels (D) to (I) in Fig 5, the vertical lines represent the changed score positions due to the removals of the collaborators of RD and NWA. The NWA values are nearly at the end of the distributions, suggesting that such magnitude of changes is rarely observable.

**The identity requirement.** We then assess our working hypothesis about the identity requirement with a focus on socio-economic identities. Some rap songs are exclusive and restrictive [74]. Lyrics depict delinquency and crime, social inequality and injustice, ethnicity, and personal introspection [75,76]. Such music is linked with artists' stigmatized identity as members of socially excluded or economically peripheral groups [77–79].

We study artists' membership in socially excluded groups by focusing on gang affiliations. NWA is known for representing Gangsta rap, and the group writes lyrics that are vividly sexist, misogynistic, and homophobic and describe the violence of ghetto life in the US [80]. The group's anti-social identity and culture may deter artists with no gang affiliations from using their root concepts. We create a dichotomous variable coded as 1 if an artist has a gang affiliation and 0 otherwise from the website of *Rappers and their Gang Affiliations*, where the web viewers report artists' gang affiliations based on the connoted lyrics. We also collect the data on gang affiliations from Genius, an online music and lyrics website [81].

In addition, we operationalize an artist's membership of economically peripheral groups with production resources they employ to create an album. We collect the data of production

resources from albums' credit sections, in which we find 867 unique roles and 25 typical roles that appear in over 5% of the albums (e.g., producers and composers). Thus, 842 roles are atypical (e.g., trombone and creative director). If artists adopt such unusual resources that ordinary artists cannot afford, they might not have identities as members of economically peripheral groups. The linear model for estimations is as follows:

$$y_i = \beta_0 + \beta_i \textit{Gang Affiliations}_i + \beta_2 \textit{Atypical Production Resources}_i + \sum_j \gamma_j x_{ji} + \varepsilon_i. \quad (14)$$

Fig 6 demonstrates the results of regression analyses for testing the identity requirement hypothesis. We use the same datasets, models, and estimation methods employed to analyze the contagion requirement, as illustrated in the regression model (2). We obtain the following findings. First, in an album, an artist with a gang affiliation is more likely to express NWA-like moods than one without such an affiliation ($\beta = 0.39$, $p = 2.2 \cdot 10^{-6}$). An artist's identification as a member of socially deviant groups increases the use of implosive root concepts.

Second, in an album, an artist with access to atypical production resources is less likely to express RD-like and NWA-like moods than one without such access. However, the size of the RD-like and NWA-like coefficients is –0.15 ($p = 2.3 \cdot 10^{-8}$) and –0.25 ($p < 2.0 \cdot 10^{-16}$), respectively, suggesting that their moods are much less like NWA than like RD. These findings suggest that the qualitative generative capacity of implosive root concepts is different from that of explosive root concepts due to the identity requirement: implosive root concepts require users to have a strong socio-economic identity, thereby reducing not only the pool of potential users but also the variety of elements to be recombined by them.

## Discussion and conclusion

### Summary of findings

Our exploratory analysis identified two root concepts in the history of rap music, which differ in their capacity to generate new combinations: artists join explosive and implosive root concepts with heterogeneous and homogeneous elements, respectively, in their cultural production. The rap artists RD and NWA presented root concepts to this field of cultural production. The former's root concept is explosive, whereas the latter's is implosive. NWA's root concept is more tacit than RD and thus, requires network contagion for knowledge transfers. NWA's root concept also requires unique socio-economic identities. The exclusiveness of the two

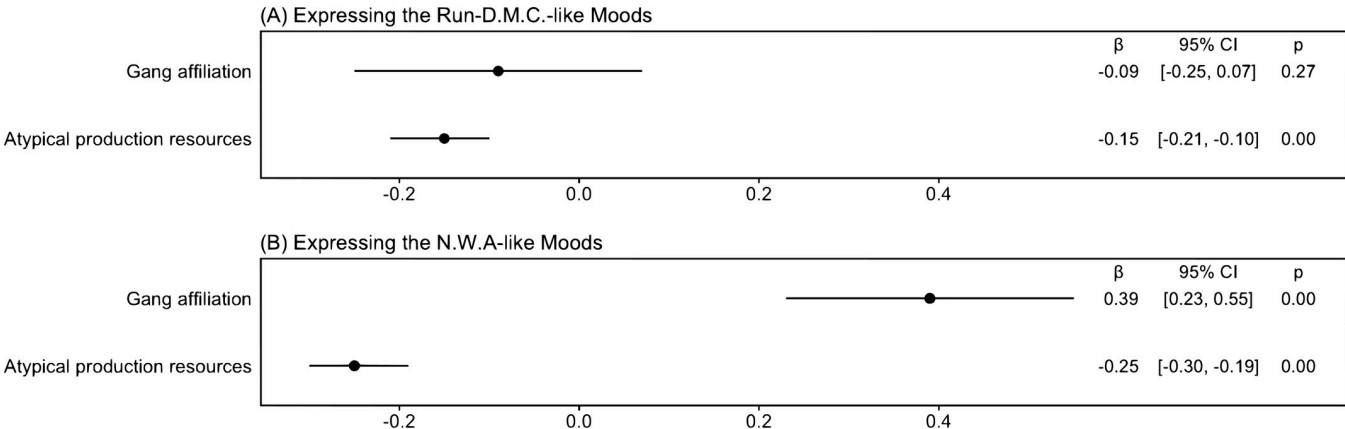

**Fig 6. Coefficient plots—In an album, an artist who has a corresponding socio-economic identity tends to express moods similar to those expressed by Run-D.M.C. or N.W.A (N = 6,111).** The figure indicates point estimates and 95% confidence intervals.

requirements limits potential users, whereby implosive root concepts are joined only with homogeneous ideas.

## Influence of great ideas

This study advances knowledge of the influence of great ideas. Fink and Reeves [30] propose and demonstrate that some elements have a greater capacity for generating more combinations than others, pointing to the quantitative aspect of great ideas in knowledge field evolution. We focus on ideas' capacity but go beyond this to consider qualitative aspects, namely, that some ideas (i.e., explosive root concepts) can generate more varied combinations than others (i.e., implosive root concepts), and we exploratively articulate sources of the differences.

This shift is possible because we introduce the construct of root concepts with the recombinatory approach. Previous research focuses on combinatory patterns between elements and highlights the benefits of (and challenges in) connecting dots remotely located in knowledge fields [82]. By contrast, the construct of root concepts led us to shift our focus to differences between various elements to be recombined with root concepts, not the distance between such elements and root concepts. In some fields in which root concepts are used more disproportionately than other ideas [4,11,23], creators can innovate by joining other elements with root concepts rather than finding valuable combinations between sheer elements.

In addition, we identify two types of root concepts: implosive and explosive. Creators use these by adding other elements, but they do so differently, depending on the type of concept. Our work corresponds to Gates et al. [29], who find that scholarly papers in some areas of study tend to self-cite, which is analogous to our notion of implosive root concepts, as some forms of knowledge are joined with homogeneous ideas. Some knowledge fields evolve by attracting various elements because root concepts in such fields are explosive. Fields in which root concepts are implosive become exclusive in the sense that only some creators are allowed to enter.

Root concepts are a powerful tool for those interested in analyzing evolutionary patterns of cultural fields. For example, one can hypothesize that some cultural fields such as ballets, operas, and country music begin with implosive root concepts and long develop upon them without introducing different elements into the fields. In such fields, creators consistently keep recombining homogeneous elements with the initial root concepts and preserve traditional styles over time. However, one can also hypothesize that other cultural fields, such as rock music, may grow extensively because of the emergence of explosive root concepts (e.g., Elvis Presley and the Beatles) and rapidly evolve by incorporating heterogeneous elements.

Moreover, another evolutionary trajectory might entail transformation from implosive to explosive root concepts. Some cultural fields, such as molecular gastronomy initiated by Ferran Adrià [83], emerged with implosive root concepts. Creators initially used them with homogeneous elements in a rather restrictive manner, but those in the subsequent generations might test and enhance root concepts' generative capacities by experimenting joint use with heterogeneous elements. This is likely when no influential gatekeeper monitors and punishes such experimentation [33]. Such transformation promotes the fields' differentiation, leading to the emergence of multiple sub-categories within the fields. Exploring this possibility requires future research to correct a static view of root concepts and presume that root concepts can dynamically change from implosive ones to explosive ones over time (or vice versa).

Future research can also reconcile this study's dichotomous view of recombination diversity. Based on our initial findings, this study focused on two extreme ends of the diversity scale (i.e., RD and NWA), but our analysis could be different if we additionally identified other influential and innovative creators with root concepts located in the middle of the scale. An

example is OutKast, one of the top artists who won Best New Rap Group at the Source Awards in 1995 and became a key player in the rise of Southern rap [84]. According to the AllMusic data, as of March 2022, the duo uses styles that have influenced trap rap and contemporary rap, and OutKast is followed by 51 artists such as J-Cole, Migos, and Rae Sremmurd. In our dataset of 1,154 rap artists, their ranking is 14th in the number of followers. Compared to RD and NWA, OutKast presents warm, sophisticated, and sleazy music moods, which the influenced artists use jointly with neither highly diversified nor highly homogeneous elements. The current restrictive view of root concepts should be relaxed to account for cases located somewhere between the two ends of the diversity scale.

## Challenges of recombination

The literature often attributes the difficulty of finding novel recombination patterns to creators' limited cognitive capacity and the resulting local search [82,85]. Cognitive costs cause them to limit their search in, for instance, the neighborhood of familiar ideas or experiences. Our findings suggest that if creators work in fields in which implosive root concepts are dominant, it is difficult for them to find novel combinations. In this case, the difficulty is not because of creators' search costs but because of the innate characteristics of root concepts (i.e., contagion and identity requirements). This new lens of explosive and implosive root concepts enables us to better understand the challenge of employing new combinations.

Our findings suggest that creators have two options when deciding on allocating resources for innovation activities: they can join other elements with explosive or implosive root concepts. Explosive root concepts confer various merits and demerits. Considering that creators can use highly diversified elements in conjunction with explosive root concepts, they can differentiate themselves by finding unique combinations. A potential drawback is that creators must incur high search and experimentation costs, given the enormous number of potential combinations. However, implosive root concepts do not entail such costs because creators can quickly undertake neighborhood searches for alternatives when the combinations prove ineffective. Given that only some creators with specific backgrounds can use implosive root concepts, this exclusiveness can be a source of differentiation. A potential downside is that they cannot make combination-level differentiation and are more likely to face problems of spanning discounts if experimental elements highly different from other elements have already been used with implosive root concepts [22].

Given that the two types of root concepts have pros and cons, how should creators astutely use them for innovation? Following Teodoridis et al. [86], we argue that in knowledge fields with fast progress wherein specialists' in-depth expertise is advantageous, creators might be better off using implosive root concepts. By contrast, in fields with slow progress wherein generalists' access to diverse knowledge sets is advantageous, the value of using explosive root concepts increases. Creators may be able to use them properly if they consider the pace of change in cultural production fields.

In addition to rigorous empirical tests of this working hypothesis, future research can explore how consumers react to creators' use of root concepts. Following Goldberg et al. [87], who find cultural omnivores who value products from multiple genres disfavor those that do not highly conform to the genres' prototypes, we can intuitively hypothesize that omnivores favor products that employ implosive root concepts in multiple domains. However, this possibility must be theorized and tested.

## Followers and protégés

Great ideas are influential because many other creators use them in their creative activities. This study contributes to the literature on the behavior of followers and protégés. Azoulay

et al. [33] find that non-protégés refrain from using and advancing the great ideas of star scientists because they face cognitive, social, and resource barriers established by star scientists and their protégés. Their findings align with those of Tzabbar and Kehoe [88]: the presence of star scientists in firms causes firm-level exploitation because they exert great influence on firms' decisions about how firms allocate resources to research themes. Moreover, their turnover allows those remaining in firms to explore and create paths in firms' knowledge trajectories. These findings suggest that protégés' loyalty to and beliefs in the creators of great ideas cause strong inertial forces in the direction of knowledge development.

This study extends the literature by proposing that the emergence of such inertial forces may depend on the types of root concepts. In particular, protégés likely receive such forces only when using implosive root concepts. By contrast, creators of great ideas might not limit potential exploration if they present explosive root concepts with which protégés can experiment with various recombinations. Therefore, future research could reevaluate previous findings using the lens of root concepts and test the arguments discussed in our study.

## Limitations

There are several limitations to our study. First, our work relies on historical data from a specific music category. The extent to which our findings are generalizable to other categories in the creative industry and other knowledge fields, such as academics and engineering, is unclear. In some knowledge fields, there might not be a root concept or various root concepts. In addition, we recognize that some of our arguments are highly specific to our empirical contexts. For example, the identity requirement for using implosive root concepts should be more meaningful in creating cultural products than that of scientific knowledge if it does not rest on specific ideologies.

Second, we view our exploratory work as illustrative or illuminative rather than definitive or decisive. We recognize that the representativeness of our samples is limited, and our findings require finer-grained analyses to test causation. Therefore, future econometric analyses are required.

Third, the extent to which the archival databases used in this study are reliable and representative is unclear. The uniqueness restricts us from conducting cross-validation or finding alternative, complementary databases, and tests with additional data in different contexts are required.

Fourth, potential competitive relationships between the two root concept creators may challenge the independence premise of this study. RD should have had an opportunity to influence markets earlier than NWA, which our model did not incorporate.

Fifth, given the limited data availability, this study did not fully incorporate subsequent creators' data, thereby limiting our exploration of between-creator differences in their use of explosive (or implosive) root concepts.

Finally, we regarded mood as an outcome of musical experience and thus used it as a phenotype of music in identifying root concepts. This focus was intentional, as we adopted the recombinatory approach to studying innovation in this genre. However, by using alternative phenotypes, such as lyrics [49], sampling [89], musical instruments [27], beats [90], or chord progression [13], future research could present further insights into root concepts' historical influence on the evolution of cultural fields and explore alternative ways of empirically capturing root concepts.

## Conclusion

In this study, we proposed the notion of root concepts to examine how they are subsequently used and why they are used differently. We believe that, despite some limitations, this work

provides exciting new avenues to study the growth trajectories of knowledge fields with a focus on root concepts and the diversity of elements to be joined with them. We hope that our work presents new opportunities for future research on the evolutionary growth of cultural production and knowledge fields.

## Supporting information

**S1 Fig. Cumulative distribution of the root concepts' influence and novelty.** To ensure that Run-D.M.C. and N.W.A are uniquely different from other artists, for the data on artists in our sample, we plotted the cumulative population-level distributions. The x-axis of panels (A) and (B) denote in-degree centrality and PageRank centrality, respectively, as influence indicators. The x-axis of panel (C) is the number of novel mood pairs as an indicator of novelty. The vertical solid and dotted lines indicate the data for Run-D.M.C. and N.W.A, respectively. They are both located at the right end of the cumulative distributions, suggesting their exceptional influence and novelty. The in-degree centrality, PageRank, and novelty scores for Run-D.M.C. are 0.998, 1.000, and 0.999, respectively, and those for N.W.A are 1.000, 0.998, and 0.994, respectively. The data suggest the two artists have extreme scores.
(TIF)

**S2 Fig. Distance between the root concepts' topic vectors (N = 19,753,756).** The Euclidean distance between Run-D.M.C.'s vector and N.W.A's vector is that its location is at the 0.90 percentile in the distribution of all possible pairs of albums' distances. The solid vertical line indicates the root's distance. The dotted line represents the mean distance.
(TIF)

**S3 Fig. An illustration of the effects of node removals.** Dots and edges represent artists and the directional influences between them. The black dots in the center are root concept creators. The gray dots are artists who work with the collaborators of root concept creators. The white dots are other artists. These diagrams illustrate changes in the overall network structures after node removal. The impact of removing nodes that work with root concept creators' collaborators is greater in network a, where root concept creators' influences and network contagions are closely related. The diagrams suggest that changes can be captured best with the reachability of the remaining nodes.
(TIF)

**S1 Table. Descriptive statistics of influence and novelty.**
(PDF)

**S2 Table. Descriptive statistics and correlations for analyzing differences in the root concepts' generative capacity (N = 4,040).**
(PDF)

**S3 Table. Regressions for analyzing differences in the root concepts' generative capacity.**
(PDF)

**S4 Table. Descriptive statistics and correlations for analyzing the contagion and identity requirements (N = 6,111).**
(PDF)

**S5 Table. Regressions for analyzing the contagion and identity requirements.**
(PDF)

**S1 File. The Zip file contains the final datasets that we used for the analyses and R codes to reproduce the results reported in the manuscript.**
(ZIP)

## Acknowledgments

We thank Kyosuke Tanaka (Aarhus University) and Makoto Mizuno (Meiji University) for their helpful comments.

## Author Contributions

**Conceptualization:** Susumu Nagayama, Hitoshi Mitsuhashi.

**Data curation:** Susumu Nagayama, Hitoshi Mitsuhashi.

**Formal analysis:** Susumu Nagayama, Hitoshi Mitsuhashi.

**Funding acquisition:** Susumu Nagayama, Hitoshi Mitsuhashi.

**Investigation:** Susumu Nagayama, Hitoshi Mitsuhashi.

**Methodology:** Susumu Nagayama, Hitoshi Mitsuhashi.

**Project administration:** Susumu Nagayama, Hitoshi Mitsuhashi.

**Resources:** Susumu Nagayama, Hitoshi Mitsuhashi.

**Software:** Susumu Nagayama.

**Supervision:** Susumu Nagayama, Hitoshi Mitsuhashi.

**Validation:** Susumu Nagayama, Hitoshi Mitsuhashi.

**Visualization:** Susumu Nagayama, Hitoshi Mitsuhashi.

**Writing – original draft:** Susumu Nagayama, Hitoshi Mitsuhashi.

**Writing – review & editing:** Susumu Nagayama, Hitoshi Mitsuhashi.

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
