## [Decision Letter · Decision Letter 0]

9 Feb 2022

PONE-D-21-29058Explosive and implosive root concepts: An analysis of two artists in rap musicPLOS ONE

Dear Dr. Nagayama,

Thank you for submitting your manuscript to PLOS ONE. After careful consideration, we feel that it has merit but does not fully meet PLOS ONE’s publication criteria as it currently stands. Therefore, we invite you to submit a revised version of the manuscript that addresses the points raised during the review process.

We look forward to receiving your revised manuscript.

Kind regards,

Jichang Zhao, Ph.D.

Academic Editor

PLOS ONE

Journal Requirements:

Reviewers' comments:

Reviewer's Responses to Questions

**Comments to the Author**

1. Is the manuscript technically sound, and do the data support the conclusions?

Reviewer #1: Yes

2. Has the statistical analysis been performed appropriately and rigorously? 

Reviewer #1: Yes

3. Have the authors made all data underlying the findings in their manuscript fully available?

Reviewer #1: No

4. Is the manuscript presented in an intelligible fashion and written in standard English?

Reviewer #1: Yes

5. Review Comments to the Author

Reviewer #1: This manuscript concentrates on an interesting topic - evolutionary growth of cultural production, which proposes the notion of root concepts and explores the role of roots in cultural evolution. The authors found implosive and explosive concepts for root concepts, and implosive possess innate characteristics that limit creators from experimentally integrating diversified elements. I appreciate that quantitative analysis of real data and in-depth discussion in the area of cultural (musical) evolution in this study.

However, it has some important issues and limitations.

1. As listed in the section of limitations, this study regards moods as outcomes of musical experience, and uses it as the prototype of music. In my opinion, music does not only this dimension, even not the most important one. The results in this manuscript only describe and analyze the dimension. Authors need emphasize it in the title and the abstract, and then discuss the relationship between musical emotion and creation of cultural products.

2. I doubt whether we could summarize two patterns (root concepts) through only two artists (rap music). Please deeply discuss it.

3. Most of quantitative approach are introduced by words, instead of symbols and formulas. This way of writing is not clear for readers. Please supplement symbols and formulas.

3. The methods in refs [4] and [58] are important. Please introduce them.

3. Please make the data public.

Minor issues:

1. A few URLs appear in the main text. Authors could cite them as refs or note them below the main text.

2. I suggest that two artists Run-D.M.C. and N.W.A. could be written in shorter names.

3. In the Fig.4, the numbers and the plots are overlapped. Please adjust it.

6. PLOS authors have the option to publish the peer review history of their article (what does this mean?). If published, this will include your full peer review and any attached files.

Reviewer #1: No

---

## [Author Response · Author response to Decision Letter 0]

2 May 2022

Reviewer 1

Thank you for your valuable comments. We have revised our manuscript according to your suggestions, and your comments helped us understand the strengths and weaknesses of our work. 

1. As listed in the section of limitations, this study regards moods as outcomes of musical experience, and uses it as the prototype of music. In my opinion, music does not only this dimension, even not the most important one. The results in this manuscript only describe and analyze the dimension. Authors need emphasize it in the title and the abstract, and then discuss the relationship between musical emotion and creation of cultural products.

We agree with this weakness of our work. Although there should be alternative ways of conceptualizing and empirically categorizing music, we focus only on music moods because this approach works well when studying innovations with the recombinatory approach. This limitation was not very explicit in the previous version, so we have made the following changes: 

- We changed the title of our paper. The new title is “Explosive and implosive root concepts: An analysis of music moods rooted by two influential rap artists.” 

- We changed the abstract by noting that “Using data on artists and albums in the rap genre from the online database Allmusic, we view music moods as a type of experience that music generates and focus on music moods as a phenotype in studying styles and modes. We constructed a dataset of recombinatory patterns in the subsequent cultural production and identified two types of root concepts.” (p. 2)

- In the introductory section, we also re-emphasized our position by noting that “In analyzing the data, we view music moods as a type of experience that music generates. Of the various analytical dimensions in studying styles and modes, we focus on music moods, which fits well with the recombinatory approach that analyzes combinations of elements” (p. 5).

- In addition, on p. 10, we also developed our argument on the view of moods as one of the key components of music by noting that: 

Of the other key components of music, such as melodies, harmonies, and rhythm, we view music moods as phenotypic elements to be recombined. Music generates moods and interactions between artists and audiences. Indeed, previous research used music moods as a phenotype in styles and modes [49,53–55]. In a seminal work, Hevner [54] studied the expressiveness of music through the analysis of affective values that music generates, such as joy, dignity, and dreaminess. Spotify also supports this view of moods as one of the key components by developing technology to draw inferences about audiences’ moods based on their voices and suggest songs that match emotional states [56]. 

- In the limitations section (p. 34), we also noted that future research should explore alternative phenotypes to study the evolution of cultural fields by taking the recombinatory approach. 

2. I doubt whether we could summarize two patterns (root concepts) through only two artists (rap music). Please deeply discuss it.

We believe that you are pointing out that our conceptualization of root concepts might be oversimplified in two ways. First, our conceptualization is static in the sense that this study makes no consideration of dynamic processes in which an implosive root concept can gradually become an explosive one over time. Second, we adopted a dichotomous view and focused only on two extreme ends of the diversity scale. 

We agree with these shortcomings in our conceptualization and therefore presented extensive arguments in the discussion section. In particular, we made two important arguments: (1) a possibility that a root concept that was initially viewed as implosive can become explosive, and (2) another possibility that a root concept can be located between the two extreme ends of the diversity scale. We illustrate (2) using the case of OutKast. Please refer to our discussion on pp. 29–30: 

3. Most of quantitative approach are introduced by words, instead of symbols and formulas. This way of writing is not clear for readers. Please supplement symbols and formulas.

We now use symbols and formulas in explaining our measurements and empirical methods. For example, we present a mathematical formula of our novelty measurement with symbols (e.g., p.13). 

4. The methods in refs [4] and [58] are important. Please introduce them.

We agree with this. As you will notice, [58] is now numbered [54]. These studies are relevant particularly to our discussion about music moods as a phenotype in studying music styles and modes. In this version, we added the following arguments on p. 10 in the section on the research context. 

Studies adopting the recombinatory approach hinge on the phenotype of knowledge. For example, in their analysis of the patent application data, Fleming et al. [4] viewed patent subclasses as phenotypes for studying combinations and measured the novelty of an innovator’s technological and intellectual knowledge with the number of new subclass combinations in each of their patent applications. As another example, in their analysis of innovation in the fashion industry, Godart et al. [52] decomposed clothing styles into color, fabric, print, pattern, and look and examined how fashion houses adopt combinations of these elements. 

5. Please make the data public.

The final datasets and program codes in R that we used for the analyses are now available at the following site: 

https://github.com/nagayaman/rootconcept?fbclid=IwAR0K7wS5_uTj3e4dtEzYNF_2YG9Yo9OxrovyDkpCM9WJhvSbYTwsG1M4BP0

Please note that the repository contains data that we have processed for the purpose of the analyses. 

1. A few URLs appear in the main text. Authors could cite them as refs or note them below the main text.

Thank you. We have fixed this reference issue. 

2. I suggest that two artists Run-D.M.C. and N.W.A. could be written in shorter names.

We now use the following abbreviations for the two artists: 

RD: Run-D.M.C. 

NWA: N.W.A

3. In the Fig.4, the numbers and the plots are overlapped. Please adjust it.

We now have adjusted them in the figure. Please note that we also have re-updated figures and tables in the manuscript and supporting files. In this re-updating process, we conducted the same analyses again using the same data. The newer version of the programming packages in R presents results slightly different from those we presented before, but our findings and conclusions remain the same.

---

## [Decision Letter · Decision Letter 1]

15 Jun 2022

Explosive and implosive root concepts: An analysis of music moods rooted by two influential rap artists

PONE-D-21-29058R1

Dear Dr. Nagayama,

We’re pleased to inform you that your manuscript has been judged scientifically suitable for publication and will be formally accepted for publication once it meets all outstanding technical requirements.

Kind regards,

Jichang Zhao, Ph.D.

Academic Editor

PLOS ONE

Additional Editor Comments (optional):

Reviewers' comments:

Reviewer's Responses to Questions

**Comments to the Author**

1. If the authors have adequately addressed your comments raised in a previous round of review and you feel that this manuscript is now acceptable for publication, you may indicate that here to bypass the “Comments to the Author” section, enter your conflict of interest statement in the “Confidential to Editor” section, and submit your "Accept" recommendation.

Reviewer #1: All comments have been addressed

2. Is the manuscript technically sound, and do the data support the conclusions?

Reviewer #1: Yes

3. Has the statistical analysis been performed appropriately and rigorously? 

Reviewer #1: Yes

4. Have the authors made all data underlying the findings in their manuscript fully available?

Reviewer #1: Yes

5. Is the manuscript presented in an intelligible fashion and written in standard English?

Reviewer #1: Yes

6. Review Comments to the Author

Reviewer #1: My comments and concerns have been addressed. In my opinion, this version of manuscript is ready for publication.

7. PLOS authors have the option to publish the peer review history of their article (what does this mean?). If published, this will include your full peer review and any attached files.

Reviewer #1: **Yes: **Zhenkun Zhou

---

## [Editor Report · Acceptance letter]

23 Jun 2022

PONE-D-21-29058R1 

Explosive and implosive root concepts: An analysis of music moods rooted by two influential rap artists 

Dear Dr. Nagayama:

I'm pleased to inform you that your manuscript has been deemed suitable for publication in PLOS ONE. Congratulations! Your manuscript is now with our production department. 

Kind regards, 

on behalf of

Professor Jichang Zhao 

Academic Editor

PLOS ONE